# Nutraceuticals and Their Contribution to Preventing Noncommunicable Diseases

**DOI:** 10.3390/foods12173262

**Published:** 2023-08-30

**Authors:** Aurora Garza-Juárez, Esther Pérez-Carrillo, Eder Ubaldo Arredondo-Espinoza, José Francisco Islas, Diego Francisco Benítez-Chao, Erandi Escamilla-García

**Affiliations:** 1Departamento de Bioquímica y Medicina Molecular, Facultad de Medicina, Universidad Autónoma de Nuevo León, Monterrey 64460, Mexico; aurora.garzajr@uanl.edu.mx (A.G.-J.);; 2Centro de Biotecnología FEMSA, Escuela de Ingeniería y Ciencias, Tecnológico de Monterrey, Monterrey 64849, Mexico; 3Laboratorio de Farmacología Molecular y Modelos Biológicos, Facultad de Ciencias Químicas, Universidad Autónoma de Nuevo León, Monterrey 66427, Mexico; 4Microbial Biotechnology Laboratory, Centro de Investigación y Desarrollo en Ciencias de la Salud, Universidad Autónoma de Nuevo León, Monterrey 64460, Mexico; 5Facultad de Odontología, Universidad Autónoma de Nuevo León, Monterrey 64460, Mexico

**Keywords:** bioactive compounds, health, inflammatory factors, noncommunicable diseases, nourishment

## Abstract

The high rate of deaths around the world from noncommunicable diseases (NCDs) (70%) is a consequence of a poor diet lacking in nutrients and is linked to lifestyle and environmental conditions that together trigger predisposing factors. NCDs have increased 9.8% of public health spending worldwide, which has been increasing since 2000. Hence, international organizations such as the WHO, the Pan American Health Organization, and the Food and Agriculture Organization of the United Nations have been developing strategic plans to implement government and economic policies to strengthen programs in favor of food security and nutrition. A systematic review is presented to document an analysis of the origin and characteristics of obesity, cardiovascular disease, chronic respiratory diseases, diabetes, and cancers affecting a large part of the world’s population. This review proposes a scientifically based report of functional foods including fruits, vegetables, grains, and plants, and how their bioactive compounds called nutraceuticals—when consumed as part of a diet—benefit in the prevention and treatment of NCDs from an early age. Multifactorial aspects of NCDs, such as culture and eating habits, are limitations to consider from the clinical, nutritional, and biochemical points of view of everyone who suffers from them.

## 1. Introduction

Ancient Greek and Asian societies postulated the concept that some foods could be injurious, whereas others have restorative capabilities. Even Hippocrates stressed that food could be our first medicine by eating a balanced diet to preserve health [1]. Homeostasis can be referred to as an essential balance among the vital functions of the body and mind [2], a state typically achieved by having a healthy lifestyle which encompasses eating a balanced diet, performing regular physical activity, and getting enough rest [3,4].

Food plays a fundamental role in the maintenance of health [5]. Particularly, the lack of a good diet leads to the development of several noncommunicable diseases (NCDs), such as cardio-metabolic dysfunction, overweight, obesity, diabetes, cancer, diabetes, etc. [6,7]. In 2004, the World Health Organization (WHO) established a strategy on diet, health, and physical activity to be implemented around the world [8], with the aim of improving global health through educational programs focused on the correct consumption of food. Among the distinct types of foods, functional foods are widely accepted as part of the human diet due to their flavors and nutritional properties, including bioactive components and essential oils [9], which are collectively regarded as nutraceuticals [10,11].

The scope of this review aims to highlight the health relevance of the nutraceuticals contained in functional foods that are involved in the nutritional prevention and treatment of NCDs. An exhaustive search was performed using electronic databases including Scopus and PubMed. The keywords used on the web were noncommunicable diseases (NCDs), nutraceuticals, and functional foods. KingDraw’s Professional Chemical Structure Editor, copyright 2009–2023, KingDraw.cn, was used to make chemical formulas.

## 2. Development of NCDs

NCDs such as cardiovascular diseases (myocardial infarctions and strokes), cancers, chronic respiratory diseases (obstructive pulmonary disease and asthma), and diabetes represent 70% of deaths globally [12]; sixteen million people die annually before the age of 70 as a result of NCDs, and 81% of these premature deaths occur in low- and medium-income countries. Cardiovascular diseases indicate the highest annual mortality worldwide, with 17.9 million deaths; the second-highest cause is cancer with 9 million deaths, followed by 3.9 million deaths caused by respiratory diseases, then diabetes in fourth place with 1.6 million deaths [12]. Particularly, a diet containing high saturated fats, sugar, and salt, with a low intake of fruits, vegetables, whole grains, cereals, and legumes, as well as little physical activity, are factors that result in NCDs such as overweight and obesity [13]. Concerning human health, a decompensation in physiological human conditions such as high blood pressure, elevated serum glucose, hyperlipidemia, and overweight and obesity caused by an unhealthy diet, as well as other vicious conditions, such as physical inactivity and tobacco and alcohol use, can be the origin of NCDs [12,13].

NCDs tend to be long term, and they result from a combination of genetic and epigenetic processes, as well as physiological, environmental, and behavioral conditions. They do not resolve spontaneously and are rarely completely cured, which generates a socioeconomic burden because of social dependency and disability [3,14]. Thus, NCDs have become a challenge for developing public health policies as they constantly go unnoticed but over time become chronic issues, affecting hundreds of millions [3,12,15,16]. Significant challenges undoubtedly include the number of affected cases, the increasing contribution of general mortality, as well as the elevated costs of hospitalization and medical treatment, including rehabilitation for those affected. According to the above, the goal is to develop a change in public health policies, which can have an impact on modifying and/or adapting the lifestyle of the population [16,17,18].

To minimize rising mortality rates, it is critical to develop national programs among all social levels to instill a healthy eating culture, particularly in countries with medium and low socioeconomic status. Because chronic illnesses, a by-product of unhealthy food consumption [19], coexists with a high life expectancy, there has been a significant increase of about 9.8% in health spending allocated to the care of these conditions, growing over the last two decades [14,20]. The cost of care for persons with chronic illnesses, along with the excessive costs of early death and incapacity, as well as treatment expenditures, have an impact on governments and communities. Direct global public policies focusing on medium- and long-term objectives are boosting NCD prevention efforts. It is feasible through measures such as promoting health and the inclusion of physical activities as part of a healthy lifestyle, with a focus on risk groups, for the population to acquire good eating habits, favoring the empowerment of all involved sectors. For example, paternal and maternal diet can influence the genetic predisposition of newborns [6,14,18,19,21]. These actions are aligned with the proposal of the “25 × 25” strategy as mentioned in the World Health Assembly in 2011, and it was incorporated into the proposals of the World Health Organization for the global action plan on NCDs from 2013 to 2020. Here, the aim is to reduce the mortality of the population by 25% for 2025 [22].

Another “Action Plan to Eliminate Trans-Fatty Acids from Industrial Production (TFA-IPs) 2020–2025” is a joint endeavor of the Pan American Health Organization (PAO) initially focused on preventing NCDs. For example, trans-fatty acids from industrial production are an avoidable factor that contributes to the burden of cardiovascular disease, the leading cause of mortality in the Americas. In this case, the rules were critical to eliminate the industrial manufacturing of these dangerous compounds [23] and their presence in industrial foods. Current priorities in the health sector include enacting and implementing regulatory policies to eliminate Partially Hydrogenated Oils (PHO) from the food supply and/or to limit Industrially Produced Trans-Fatty Acids (IP-TFA) content (~2%) of total fat in all food products, as well as to assess the progress of those regulatory policies on the food supply and on human consumption and create awareness of the negative health impacts of Trans Fatty Acids (TFA) and the health benefits to be gained from the elimination of IP-TFA among policymakers, producers, suppliers, and the public. Therefore, an adequate healthcare system can provide an immediate and sustained response to mitigate future consequences. The government’s key influential political and economic leadership is required to enable enough public spending on health and programs [24].

### 2.1. Molecular Mechanisms in NCDs Illustrated by Overweight and Obese Stages

NCDs are characterized by presenting an inflammatory state at the cellular and organic level, influenced by lifestyle diets rich in sugars, refined flours, and saturated fats as they related to a developing overweight and/or obese stage [25]. One way to effectively address the growing problem of overweight and obesity is to improve health by focusing on access to nutritious foods to maintain healthy growth into adulthood and throughout the life of the individual [26].

In overweight and obesity stages, the hypertrophy of adipose tissue is characteristic. On molecular terms, macrophages and T cell infiltration processes are expected to happen with the production of local pro-inflammatory mediators including cytokines, tumor necrosis factor-alpha (TNF-α), interleukin-6 (IL-6), and interleukin-1b (IL-1b), as shown in Figure 1. They obstruct the insulin signaling cascade, inducing insulin resistance, and further decomposing glucose and lipid metabolism in adipose tissue, skeletal muscle, and the liver [27]. Therefore, the overweight as well as the obesity stage trigger chronic low-grade inflammation in several organs and increase cardiac metabolism. Hence, the developing dysfunction of patho-physiologically linked with other risk factors such as insulin resistance, arterial hypertension, dyslipidemia, and fatty liver disease, increasing developing Type 2 Diabetes (DM2) and atherosclerosis, which are also types of NCDs.

The survival and constant defense of human health depend on the proper metabolic and immunological functioning, which are linked and work interdependently [28]. For example, the organism depends on its innate capacity and response to repair damage when infections arise, but it can also store energy when required. Inflammation is the central reaction of the non-specific natural ability or immunity—a local response to cellular damage characterized by increased blood flow, capillary dilation, leukocyte infiltration, and localized production of chemical mediators that start the attack on harmful agents. This is then followed by the release of anti-inflammatory cytokines, inhibition of proinflammatory signaling cascades, and regulatory cells [27,29,30]. Recently, macrophages progressively infiltrate obese adipose tissue, becoming critical mediators of insulin resistance. These are commonly referred to as activated macrophages (M1), and they both secrete cytokines, e.g., interleukin-1 (IL-1) and interleukin-6 (IL-6), in addition to tumor necrosis factor-alpha (TNF-α). At least, they can create a pro-inflammatory environment blocking insulin action. In lean individuals, macrophages are activated in the (M2) state and based on wound healing and immunoregulation; they produce the cytokine IL-10, which protects against inflammation [31]. The role of Cytotoxic T cells in adipose tissue is involved in the differentiation, activation, and subsequent migration of macrophages. T cell-derived cytokines such as gamma interferon promote the recruitment and activation of M1 macrophages, increasing inflammation and insulin resistance in adipose tissue. Through dietary intervention, controlling adipose tissue T cell and macrophage activity in vivo can mitigate inflammation and restore immunological balance, representing a therapeutic aim that looks properly controlled as inflammation is essential for maintaining health and homeostasis; inflammation that involves losing regulatory processes becomes pathological and harmful [29].

In a comparable manner, the metabolic state plays a crucial role since it can modify the immune capacity of the organism to fight infections with the inflammatory response that alters the organism’s metabolism, favoring or suppressing the insulin signaling pathway. Maintaining a healthy weight means keeping the immune system in balance. In the opposite case, obesity causes a metabolic imbalance that favors immunosuppression, causing malnutrition and generating a low-grade chronic inflammatory state [32].

Exceeding the proliferative capacity of adipocytes results in hypertrophy and subsequent hyperplasia, increasing the mass of adipose tissue containing adipocytes and non-adipocytes (endothelial cells, fibroblasts, preadipocytes, leukocytes, and macrophages), which form blood vessels stroma [28]. Non-adipocytes contribute to the development of the chronic inflammatory response seen in obesity, and both adipocytes are stimulated via extrinsic or intrinsic signals [28].

### 2.2. Relation of Junk Foods Consumption with NCDs Development

Naturally, food digestion, in addition to the fat conversion into energy and the absorption of drugs and alcohols, produces free radicals in the human body. These free radicals give rise to adverse chain reactions, disturbing the cell membrane and blocking some enzymes necessary for cellular division, for example, the destruction of DNA and blockage of energy generation (oxidative stress) associated with NCDs such as diabetes, chronic disorders, and even cancer [33].

Unfortunately, nowadays, consumption of foods produced only to satisfy emotional and sensory aspects of feeding is on the rise due to the modern world’s style and pace of life. Those foods with high sugar or salt content, in addition to additives (flavors, dyes, emulsifiers, and preservatives), play a fundamental role in the palate’s sensitivity and perception of the quality of food (palatability) [34,35]. These junk foods encompass more than just carbonated drinks, juices, potato chips, candy, and industrial baked goods. Studies conducted in the United States, Canada, and Brazil consistently showed that high consumption of these junk foods brings with it an unbalanced diet, being clear that they are a constant health hazard due to their association with obesity, dyslipidemia, and even breast cancer, to name a few related problems [34,35]. Foods with a high glycemic index (carbohydrates and refined sugars) produce a critical metabolic effect on health, contributing to overweight and abdominal obesity, particularly in women [34,35].

A study conducted by scientists from Nestle Research Center and the Nestle Nutrition Global R&D [36] demonstrated the difference that exists in the consumption of complementary foods and as part of the diet of 3103 children between 6 and 23 months of age from China (n = 906), the United States of America (USA) (n = 1430), and Mexico (n = 906). Supplementary foods are necessary after the breastfeeding period, as these natural foods of plant and animal origin contribute to the child’s development and growth. The results obtained reflect the cultural aspect and eating habits of each country involved in this study. In the USA and Mexico, large portions of food ingested by children from an early age, including sugary fruit-based drinks, processed drinks, and sweets, with a low consumption of iron-fortified cereal in the case of Mexican children. These excesses are the key to promoting childhood obesity and hypertension problems, as well as diabetes [36]. Concerning China, the high consumption of refined rice, a food low in nutrients, predominates. These aspects prompted the members of the Food and Agriculture Organization of the United Nations (FAO) to analyze a sustainable food system that included economic, social, and environmental aspects to ensure food and nutritional security in the general population, which will avoid health risks and safeguard the well-being of future generations [37].

## 3. Nutraceuticals: Classification and Biological Potential

The term “nutraceutical” was coined into existence in 1989 by Dr. Stephen DeFelice from the word nutrition and pharmaceutical, and it refers to the food or part of food that provides health benefits, including prevention and treatment of diseases beyond its nutritional functions [38]—how a biologically or pharmacologically active substance (drug) alters a living organism, influencing the synthesis of proteins and the genetic material (DNA), translates into the mechanism of action [39]. So, what is the distinction between a drug and a nutraceutical? The primary objective of a drug is to address conditions with curative possibilities, and its fundamental origin is from natural medicinal herbs, manufactured by chemical, computer-aided drug design techniques or biological origins [40].

Sometimes, a drug acts on the intended targets in testing and design, contemplating toxicity, with additional benefits or side effects derived from binding to unproven targets [41]. “Nutraceuticals” are assigned to those bioactive compounds having a physiological effect intended to prevent and treat diseases or disorders other than deficiency conditions [42]. Salicylic acid, a compound originating in the white willow bark, is one of the most renowned, having been employed by the Phoenicians and ancient Greeks as an antiseptic and analgesic two millennia ago [43]. Since then, various properties of salicylic acid have been discovered, benefiting health as an antipyretic, anti-inflammatory, antibacterial, and anti-acne agent, including keratolytic and hemolytic properties that allow exfoliation of the skin where an accumulation of necrotic tissue is abundant, stimulating the regeneration of keratinocytes of the epidermis as the uppermost layer of the skin [43,44].

The biological contribution of nutraceuticals is extensive and includes the prevention or treatment of diseases in humans and animals and multiple benefits that help strengthen health. Therefore, nutraceuticals can be studied or classified from different angles to better understand their functioning in the body and the source that contains them. One classification includes three physiological stages that may limit its oral bioavailability [45]. Namely, (I) bio-accessibility (release, solubilization, and interactions), (II) absorption (mucous layer, close-junction transport, bilayer permeability, active transporters, and affluent transporters), and (III) transformation (degradation and metabolism).

There are other reports published [46,47], in which the authors present a classification of nutraceuticals from different perspectives:

(a) The source of origin: plants (garlic, aloe vera, ginger, containing organic acids, salts, tannins, and hormones), animals (oils and proteins), microorganisms (proteins, amino acids, vitamins, probiotics, prebiotics, dietary supplements, and peptides). (b) Traditional nutraceuticals: fruits (phenolic compounds, tannins, vitamins, and terpenes), vegetables (minerals, vitamins, organic acids, carotenes, and terpenes). (c) Non-traditional nutraceuticals: fortified foods (juices, cereals, vitamin additives, and minerals), originated by biotechnological processes (bread; alcoholic beverages such as beer, wine, pulque, tequila, mezcal, and apple or cane vinegar; production of amino acids and other bioceutical derivatives), or recombinant genetic engineering (production of enzymes and obtaining new nutraceutical pathways). (d) Chemical nature: herbs or spices, nutrients, phytochemicals, enzymes, terpenes (vegetables, fruits, and citrus fruits), phenolic compounds (coffee, spices, seeds, pulp and bunches of grapes, cocoa, red fruits, tea leaves, mango, banana, and spinach), and minerals (legumes, vegetables, some fruits, and spices). Nutraceuticals such as omega-3 are reported to have more than one mechanism of action. (e) The mechanism of action: their principle is to maintain and improve the physiological properties of an organism, and they are used in specific medical conditions with varied effects such as anti-inflammatory (ginger and orange peel extract), antimicrobial, osteoprotective, anti-glycemic/antihypertensive (blueberries), antioxidant (broccoli extracts), and anti-hypercholesterolemic (as the β-glucan contained in oats) [48]; however, in some cases their toxicity, and synergy, or competition are still unknown. This fatty acid has an anti-inflammatory, anticoagulant, and antithrombotic activity [49]. It is naturally in oily fish, shellfish, grains such as soy, canola, and flaxseed, walnut, to name a few.

Some widely studied bioactive compounds that have been shown to have anti-inflammatory, neuroprotective, and antioxidant activity, or as cofactors in important metabolic pathways, are presented in Table 1. Here, the information is directed to those nutraceutical compounds that have a biological contribution to health, to the foods that contain them, and to therapeutic use, helping in the prevention or treatment of diseases, according to reported data.

### 3.1. Vitamins

*Vitamin C.* This vitamin represents a redox system comprising 2 L-isomers: ascorbic acid (vitamin C) in the reduced state and dehydro-ascorbic acid (DHA) in the oxidized form. Most of its functionality in the human body is related to its role as an electron donor; hence, it is active and is stable in tissues. It is used as a cofactor or antioxidant; it is oxidized and becomes unstable [112]. Vitamin C intake as part of one’s diet has a positive effect on many illnesses such as the common cold, cardiovascular illnesses, and certain types of cancer, in addition to age-related macular degeneration, cataracts, diabetes, rheumatoid arthritis, and even has a protective effect on periodontal tissues, reducing severe cases of periodontitis [113]. For many biosynthetic and gene regulatory enzymes, this vitamin plays a crucial role in immunomodulation because it stimulates neutrophil migration to the site of infection and enhances phagocytosis and oxidant generation [55,112].

In contrast with many other vitamins, the content of vitamin C in various foods is high (10–100 mg/100 g), in some cases, reaching units of grams per 100 g of fresh weight. Most people get much of their daily vitamin C intake through regular fresh fruits and fruit juices, as illustrated in Figure 2 [112].

*Vitamin D*: The classical function, which involves mineral balance and skeletal maintenance, has been known for many years. With the discovery of vitamin receptors in various tissues, several other biological functions of this vitamin are recognized today. Its activity is explored in several human disorders such as cancer, diabetes, hypertension, cardiovascular disease, and immunological and dermatological alterations. Vitamin D occurs in two primary forms: vitamin D2 or ergocalciferol synthesized by ultraviolet B (UV-B) irradiation of the ergosterol contained in yeast, fungi, and a few natural foods, fortified food, and supplements. Vitamin D3 or cholecalciferol is obtained through a photochemical reaction in the skin and diet via the intake of animal-based foods (cod liver oils and oily fish) [114].

*Vitamin E* (*α-tocopherol*): It has been recognized as an essential lipophilic antioxidant in humans protecting lipoproteins, polyunsaturated fatty acid (PUFA), and cellular and intracellular membranes from damage for a long time [50]. One of the more interesting effects of this nutraceutical is its use in the treatment of Alzheimer’s disease. Vitamin E’s tocopherol and tocotrienol isoforms have multiple properties, including potent antioxidant and anti-inflammatory characteristics and influences on immune function, cellular signaling, and lowering cholesterol. In general, nuts and vegetable oils are rich in tocopherols, whereas barley, oat, palm oil, rice bran, rye, and wheat germ are rich in tocotrienol. In addition, natural sources of both vitamin E isomers are also in other daily foods, i.e., fruits, seafood, cheese, and eggs.

*Retinoic acid* (*RA*): RA is a metabolite made from vitamin A and is the driving force behind many of vitamin A’s characteristics. It has been documented that its three isomers, 9-cis, 13-cis, and all-trans RA, have diverse capabilities to modulate cellular growth and differentiation. Other abilities of these isomers to regulate cellular growth and differentiation may be attributable to differences in affinities to their nuclear receptors [115].

### 3.2. Minerals

*Selenium*: It is most easily absorbed in organic compounds and in the presence of vitamins A, D, and E. The primary sources of selenium in the diet are foods such as cereals, meat, dairy products, seafood, and nuts (Figure 2). Rich sources of selenium are sea salt, eggs (only in the case of Se-yeast supplementation of feed), giblets, yeast (containing selenium), bread, mushrooms, garlic, asparagus, and kohlrabi (enriched with this element) [116].

*Zinc*: Among the various food products, red meat, some seafoods, dairy products, nuts, seeds, dried legumes, and whole-grain cereals are considered good dietary sources of zinc. It is an essential trace element or a micronutrient essential for the growth and reproduction of all higher plants, animals, and humans. It is vital for the functionality of over 300 enzymes, stabilization of DNA, and gene expression. In addition, it plays a key role during physiological development and fulfils an immune function. The properties of zinc, such as toxicity, teratogenicity, carcinogenicity, and immunological, are relevant to cancer prevention [117].

### 3.3. Plants

*Silymarin*: This compound is derived from the leaves of milk thistle (*Silybum marianum*), Figure 2. The main bioactive components of silymarin are flavonolignans: silybin, silicristin, silidianin, isosilibin, dehydrosilibin, and some flavonoids, taxifolin [3]. The mixture of silybin A and silybin B (1:1) is the main active ingredient (approximately 50%) of silymarin. Silymarin flavonolignans are agents that typically possess antioxidant, anti-inflammatory, immunomodulatory, and hepatoprotective properties [59,80]. Consequently, silymarin has been suggested to be incorporated as a complementary treatment for inflammatory liver diseases including cirrhosis, hepatitis, alcoholic fatty liver disease, and non-alcoholic fatty liver disease (NAFLD) [59]. It is also known as a neuroprotective agent against many neurological disorders including Alzheimer’s and Parkinson’s diseases and cerebral ischemia. Recent studies reported its antiviral activity against various viruses, including flaviviruses (from hepatitis C and dengue), togavirus (i.e., Chikungunya and Mayaro), influenza virus, human immunodeficiency virus, and that of hepatitis B [80].

*Epicatechin*: Epicatechin is a flavonoid polyphenol from the flavanol group; it is a natural component in cocoa and its products, found in dark chocolate and green tea. Scientific evidence has shown that populations that consume substantial amounts of cocoa daily (Figure 2) have low blood pressure and a significantly lower incidence of cardiovascular disease [82,83]. The literature proposes that one of the main beneficial effects of epicatechin is achieved through its ability to remove reactive oxygen species (ROS) directly or indirectly by reacting chemically with ROS or by modulating pathways that regulate ROS-scavenging compounds and enzymes, respectively [81,83].

Epicatechin can also combat diseases such as diabetes mellitus and cancer with an inflammatory component. Epicatechin interferes with cancer signaling, making cells more susceptible to apoptosis, an effect that could be used to sensitize cancer cells to radiation or chemotherapy treatment. In addition, epicatechin appears to inhibit the proliferation of Hodgkin lymphoma cells and Jurkat T cells by inhibiting the binding of NF-κB to DNA in these cells [82,83].

*Curcumin*: A yellow pigment is the main component of turmeric, the rhizome of *Curcuma longa* XU [101]. At the molecular level, it exhibits anti-inflammatory activity through the suppression of various cellular signaling pathways, including NF-κB, STAT3, Nrf2, ROS, and COX-2. Numerous studies have indicated that curcumin is a very potent antimicrobial agent and is active against various chronic diseases, including multiple types of cancers, diabetes, obesity, in addition to cardiovascular, lung, neurological, and autoimmune diseases [84,86].

*Resveratrol*: 3,5,4′-trihydroxystilbene is a naturally occurring phytochemical. Its trans-isomer glycosylated form is found in plants as *Polygonum cuspidatum*, fruits including grapes and berries, peanuts, and red wine, Figure 2. Its use as a nutraceutical has been studied in both animal and human models, including clinical trials, in people with obesity, metabolic syndrome, heart disease, and cancer, among others; however, to date, there are no specific recommendations on the dose and duration of supplementation [59,91]. Resveratrol is a compound with antioxidative properties and is considered for treatment of neurodegenerative diseases characterized by elevated levels of oxidative damage in, for example, Alzheimer’s, Parkinson’s, and amyotrophic lateral sclerosis [92].

Previously mentioned, resveratrol may improve or assist in the treatment of metabolic syndrome. There is scientific evidence of the anti-inflammatory effect of resveratrol. This compound inhibits the production of pro-inflammatory cytokines and the activity of cyclo-oxygenases (COX-1 and COX-2) and inducible nitric oxide synthase [89,91].

*Oroxylin A*: This is a natural flavonoid isolated from *Scutellaria baicalensis* and widely used in traditional oriental medicine. It exhibits several beneficial effects, including anti-inflammatory, anti-cancer, antiviral, and antibacterial activity [93,94,95,98,99]. Oroxylin A has exerted inhibition of cell growth and induces apoptosis in various cancer cells. Oroxylin A can help treat colitis, allergy, and liver damage; these anti-inflammatory effects are attributed to the regulation of the NF-κB regulatory pathway with the consequent recruitment of immune cells and release of cytokines [94,97,99].

*Quercetin*: Another flavonoid present in fruits, vegetables, tea, and red wine (Figure 2), quercetin has shown beneficial health effects as an antioxidant and having anti-inflammatory, anti-allergic, and anti-thrombotic effects, with vasodilatory actions [101,102]. Quercetin reduced systolic blood pressure and plasma-oxidized LDL concentrations, evidencing that quercetin may safely protect against cardiovascular disease [100]. Due to its biological properties, quercetin was tested for treatment of neurodegenerative diseases, by evaluating the effect on signaling events of dopaminergic neuronal models and further testing its efficacy in the MitoPark transgenic mouse model of Parkinson’s disease [103].

*Apigenin*: It is abundant in fruits and vegetables; it has antioxidants, anti-inflammatory, and anti-cancer effects. It exerts a protective effect against oxidative stress-related diseases, including metabolic disorders, obesity, neurodegenerative diseases, and cancer [83,108,109,110]. Obesity represents a significant health issue; the administration of apigenin in obese mice increased levels of NAD^+^ coenzyme, causing a global decrease in the acetylation of proteins, improving the homeostasis of glucose and lipids [111]. Also, the use of apigenin against colon cancer shows a time- and dose-dependent cell cycle arrest in the G2/M phase in colon carcinoma cell lines [83].

*Genistein*: Obtained from soy-based foods, genistein is a phytoestrogen compound that can bind to estrogen receptors [104,105]. Besides cancer treatment, biological potential in cognitive function has also been studied in cardiovascular and skeletal health. Genistein has neuroprotective and memory-enhancing effects. Its potential is attributed to antioxidant, anti-inflammatory, and cholinesterase inhibitory activities. Genistein helps treat breast cancer due to its estrogen receptor antagonist activity, induces apoptosis, and promotes synergistic inhibitory effects when combined with anticancer drugs [105].

## 4. Conclusions

The information reviewed in the scientific literature and databases of world health authorities allowed us to perceive a common concern towards preventing noncommunicable diseases from childhood and at all stages of life. It is important to acquire information about natural food sources and food supplements, as they are a source of beneficial nutrients that sustain health. The campaigns of dissemination and teachings of a healthy diet aimed at the population have their origin in the regulatory body of health in each country and by health professionals, but it is undoubtedly crucial that in these educational campaigns, parents, or guardians become aware of a healthy diet from an early age in their children because they have the responsibility to apply and transmit that message of the food and nutritional culture as a part of comprehensive education. The home is where natural foods necessary to ensure the functioning and permanence of the microbiota must exist daily. The prevention of noncommunicable diseases in the family’s smallest members is undoubtedly the result of the example given because children learn and imitate what they observe in their environment.

Disease prevention must be paramount to ensure quality of life. Consuming natural, nutritious foods is important, but the biggest challenges lie in the socio-cultural and economic aspects of each country. Lacking access to nutritious and quality food, people who suffer from hunger are underperforming in education and work, affecting their quality of life. The same is true for the socially responsible food industry and population, as malnutrition is not just a matter of ignorance but of marketing and easy access to processed foods. Around the world, while steps are being taken, there is still much work to be done. It is the responsibility of each of us as individuals to reinforce these healthy eating strategies from childhood and make them part of our lives, and of course, they are passed down from generation to generation. With the information collected in this article, we realize that the consumption of plants in the diet, as a varied intake of fruits, vegetables, whole grains, and legumes that are rich in essential nutrients or bioactive compounds with nutraceutical characteristics, help the maintenance of healthy and strong tissues, as well as the decrease in inflammation factors and the prevention not only of infectious diseases but of NCDs as has been reviewed here. But also on the other hand, much work remains to be done with scientific evidence to establish data that specify the caloric intake, the dose, and the best way to consume each one of the bioactive compounds looking for a health benefit and considering multifactorial characteristics of NCDs as well as individual needs (clinical, biochemical, and nutritional aspects), of patients suffering from these conditions. In short, health education and promotion, environmental change, and regulation are important drivers of public health policy.

## Figures and Tables

**Figure 1 foods-12-03262-f001:**
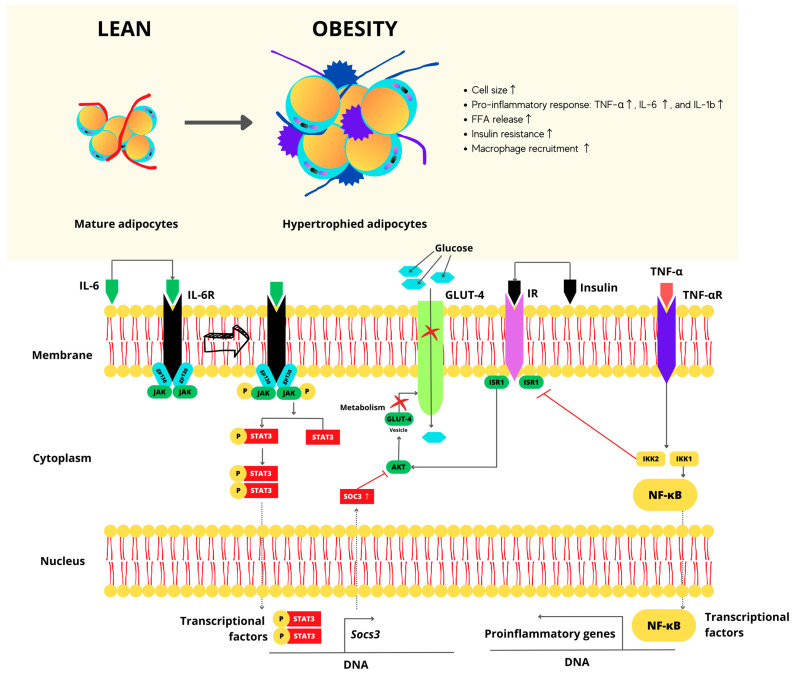
Molecular perspective for hypertrophy of adipose cells. Arrow: indicates the connection of the IL-6R to its receptor and its activation. Red cross: indicates blockages in transcription and gene expression agent, thus repressing the expression of the GLUT-4 receptor and consequently preventing the entry of glucose.

**Figure 2 foods-12-03262-f002:**
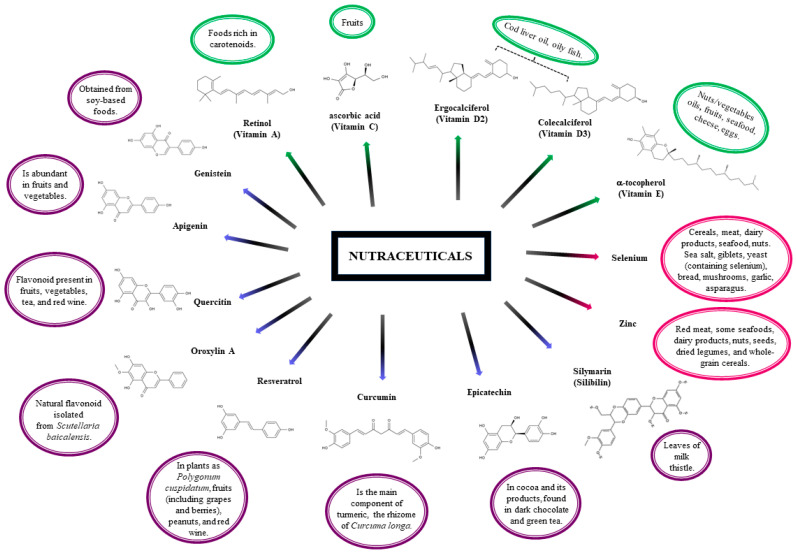
Chemical formulas for nutraceutical compounds such as Vitamins (
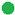
), Minerals (
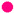
), and Plants (
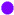
) lie to dietary source.

**Table 1 foods-12-03262-t001:** Bioactive compounds with nutraceutical characteristics and their impact on health.

Nutraceutical	Biological Activity	Source	Potential Therapeutics
Vitamin E (α-tocopherol)	Lipophilic antioxidant, protection of lipoproteins, PUFA, cellular and intra-cellular membranes from damage [50]. Neuroprotector has anti-inflammatory and hypocholesterolemic properties [51].	Avocado, banana, tomato, plums, broccoli, spinach, grains, nuts, seeds, and vegetable oils [50,52,53].	Alzheimer’s disease [51,52], cancer [53], liver disorders [54].
Vitamin C	Acting as an antioxidant, it plays a role in detoxification processes. It participates as an enzymatic cofactor and modulates synaptic activity and neuronal metabolism, among other functions [55,56].	Tapioca, beet, oranges, other fruits, cabbage, tomato, and corn [57,58].	Neurodegenerative diseases [56], cancer [58], pain [55], liver diseases [59], infections [57], SARS-CoV-2 infection [60,61,62].
Vitamin D *	Influences the immune response by modulating several immune pathways, inhibiting T cell proliferation as well as interleukin (IL)-17 and interferon (IFN)-γ [62,63].	Juices, cereals, UV-exposed mushrooms, meat, poultry, and fish [64]. * *Exposure to natural sunlight*.	Cystic fibrosis [63], colorectal cancer [65], Diabetes mellitus type 2 [66], cardiovascular diseases [67].
Retinoic acid	Inhibition of the NOTCH1 pathway [68]. A potent cell differentiation factor [69].	Leafy greens, carrots, cantaloupe, and liver [70].	Breast cancer with HER2-positive [68], neurodegenerative diseases [69].
Selenium	Intake at the recommended level assures the balanced expression of bioactive selenoproteins, which act as oxidoreductases, redox signal regulators, or thyroid hormone activation [71].	Vegetables, meat, and fish [72].	Cardiovascular diseases, cancer, and immune disorders [71], SARS-CoV-2 infection and another infection [62,72].
Zinc	Regulates the innate and adaptive immune responses and has several antioxidant effects [73].	Legumes, fortified cereals, whole grains, red meat, and some shellfish [73].	Several infections, neurodegenerative diseases [73], cancer [74], diabetes mellitus [75], cardiovascular diseases [76].
Silymarin **	It reduces inflammation and fibrogenesis, stimulates liver regeneration, exerts membrane-stabilizing and antioxidant activity [59]. Induces growth inhibition and apoptosis by suppressing the MAPK signaling pathway [77].	Fruits and seeds of the milk thistle plant (*Silybum marianum* L.) [77].	Liver diseases [59], human gastric cancer KIM 2019 [77], inflammation [78], neurodegenerative diseases [79], viral infections [80].
Epicatechin	Antioxidant activity reduces inflammation, improves endothelial function and insulin resistance [81].	Grapes, berries, apples, beans, peas, tea, and cocoa products [82].	Cardiovascular diseases [81], cancer [83], diabetes mellitus [82].
Curcumin	An antioxidant, this multitargeted agent has been shown to exhibit anti-inflammatory activity by suppressing numerous cells signaling pathways, including NF-κB, STAT3, Nrf2, ROS, and COX-2 [84]. Reduced Notch-1 activation, expression of Jagged-1 and its downstream target Hes-1 [85].	Rhizome of *Curcuma longa* [84,86].	Esophageal cancer [85], inflammatory and neurodegenerative diseases, diabetes mellitus [84], liver diseases [86], SARS-CoV-2 infection [87,88].
Resveratrol	Possesses antioxidant, anti-inflammatory, cardioprotective, and anticancer properties [89]. Induces an increase in the levels of the pro-apoptotic proteins p53, its effector p21waf and Bax [90].	Peanuts, grapes, pines, berries, red wine and *Polygonum cuspidatum* [89,91].	Leukemia [90], cardiovascular diseases, SARS-CoV-2 infection [88], cancer [89], Metabolic syndrome [91], Alzheimer’s [92].
Oroxylin A	Anti-inflammatory and anticancer properties. Reduced Notch-1 activation [93] inhibited glycolysis [94].	*Scutellaria radix* and the root of *Scutellaria baicalensis* [93].	Breast cancer [93,94], viral infections [95], liver diseases [96], colitis [97], allergy [98], neurodegenerative diseases [99].
Quercetin	Antioxidant, anti-inflammatory, antithrombotic and vasodilatory actions. Decreased plasma concentrations of atherogenic oxidized LDL [100,101].	Onions, broccoli, apples, berry crops, grapes, tea, and wine [100,102].	Cardiovascular diseases [100], cancer [101], allergy [102,103], neurodegenerative diseases [103].
Genistein	Antioxidant and anti-inflammatory activity [104]. Acts in several signaling pathways [105].	Fruit, nuts, vegetables, legumes, and soy products [105].	Cancer [83,105], viral infections [106], Huntington’s disease [104], diabetes mellitus [107].
Apigenin	Anti-inflammatory and anticancer properties [108].	Fruits and vegetables [108].	Cancer [83], viral infections [108], Alzheimer’s [109], obesity [110], metabolic syndrome [111].

* Includes Vitamin D2 (Ergocalciferol) and Vitamin D3 (Colecalciferol). ** Contains *Silibilin* as active compound.

## Data Availability

Not applicable.

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
