# Peer review of "Nutraceuticals and Their Contribution to Preventing Noncommunicable Diseases"

_foods, 2023, doi:10.3390/foods12173262_

Round 1

Reviewer 1 Report

The concept of the review is interesting about the unprocessed waste-based bioactive compounds to control non-communicable diseases. But the review is not in a compressive manner. The following comments can help the authors improve the manuscript.

In the abstract, the authors should better highlight the outcomes of the review instead of talking mainly about the research topic

Provide recent WHO data instead of 2004

Structurally, the review format also seems confusing to me, since it needs to properly organize as per the main theme of this work

 A review should include also a more critical approach to current knowledge and current experimental focus and design.

Also a clearer statement of what distinguishes this review from others; what is new here?

The article does not provide a detailed discussion on other Non-communicable diseases (NCDs), which include heart disease, stroke, cancer, diabetes, and chronic lung disease, which account for 74% of all deaths worldwide. The author selected only a few.

While mentioning the abbreviations in the manuscript, mention the first letter as Capital like "Linear Programming (LP)" instead "linear programming (LP)". Corrections are needed throughout the paper.

Recent advancements in the field of study should be included in the literature. It would be preferable if you listed the significant literature review in a table form.

The paper lacks proper sections, section titles, and connections between sections.

The section and subsections need to be clearly identified.

The authors may add commercially available bioactive substances from unprocessed foods. if any.

The authors may add the source (unprocessed food) bioactive substance in Table 1

The author may add an additional informative table to strengthen the manuscript's quality.

Briefly discuss the advancement in compound extraction from unprocessed food in a separate section.

The discussions lack critical evaluations, this is a serious issue that needs to be considered carefully throughout the manuscript.

Furthermore, I recommend that a conclusive section be added to the manuscript along with recommendations for future studies.

Finally, careful proofreading in English should be carried out. Some sentences are difficult to understand.

Revisit the entire manuscript for grammar corrections. 

Author Response

Dear reviewer, all comments were taken care of and are shown point by point.

Responses to the referees’ comments.

REVIEWER 1

  1. The concept of the review is interesting about the unprocessed waste-based bioactive compounds to control non-communicable diseases. But the review is not in a compressive manner. The following comments can help the authors improve the manuscript.
  2. In the abstract, the authors should better highlight the outcomes of the review instead of talking mainly about the research topic. The abstract was adjusted according to the content of the manuscript.
  3. Provide recent WHO data instead of 2004 A historical adjustment was made to the wording.
  4. Structurally, the review format also seems confusing to me, since it needs to properly organize as per the main theme of this work. The structure of the manuscript was revised and adjusted, also the subtopics were reorganized and reordered.
  5. Areview should include also a more critical approach to current knowledge and current experimental focus and design. In the section of the conclusion, a current critical approach was modified and applied, as well as the projection for the future.
  6. Also a clearer statement of what distinguishes this review from others; what is new here? The novelty in this document lies in combining in a structured line, the epidemiological data on NCDs diseases, the molecular mechanism of the inflammatory process at the cellular level, and the main nutraceutical compounds contained in foods of plant origin, grains, and plants; that can prevent and contribute to the treatment of NCDs.
  7. The article does not provide a detailed discussion on other Non-communicable diseases (NCDs), which include heart disease, stroke, cancer, diabetes, and chronic lung disease, which account for 74% of all deaths worldwide. The author selected only a few. The objective of this article is to address NCDs in a general way, and to mention those that are the main cause of death worldwide, since they represent a significant economic burden on health systems.
  8. While mentioning the abbreviations in the manuscript, mention the first letter as Capital like "Linear Programming (LP)" instead "linear programming (LP)". Corrections are needed throughout the paper. Corrections were addressed.
  9. Recent advancements in the field of study should be included in the literature. It would be preferable if you listed the significant literature review in a table form. Suggestions were addressed.
  10. The paper lacks proper sections, section titles, and connections between sections. Recommendations were addressed by adding and/or adjusting titles and sub-themes.
  11. The section and subsections need to be clearly identified. it has been done!
  12. The authors may add commercially available bioactive substances from unprocessed foods. if any. La observación solicitada consideramos no va acorde con los objetivos planteados en el artículo, y no se puede interponer con un conflicto de intereses.
  13. The authors may add the source (unprocessed food) bioactive substance in Table 1 A column was added to table 1, corresponding to the source of the bioactive substances; as well as the respective references.
  14. The author may add an additional informative table to strengthen the manuscript's quality. The suggestion was followed.
  15. Briefly discuss the advancement in compound extraction from unprocessed food in a separate section. We consider the information found i the current manuscript to be sufficient, besides the extraction of compounds from unprocessed foods, is not the focus of this article.
  16. The discussions lack critical evaluations, this is a serious issue that needs to be considered carefully throughout the manuscript. The suggestion was generally addressed in the different sections of the manuscript, highlighting a critical discussion.
  17. Furthermore, I recommend that a conclusive section be added to the manuscript along with recommendations for future studies. This recommendation was addressed in the conclusion section, adding a recommendation of the areas to be explored in future studies.
  18. Finally, careful proofreading in English should be carried out. Some sentences are difficult to understand. The observations of corrections and adjustments were attended throughout the manuscript.

 Comments on the Quality of English Language:  Revisit the entire manuscript for grammar corrections. It has been done.

Reviewer 2 Report

Dear Authors

The MS entitled “Bioactive chemicals in unprocessed foods and how they help prevent non-communicable diseases.” Was reviewed carefully. The MS describes the main health issues associated with poor diets and lacking bioactive compounds. Moreover, they also wrote about the nutritional alternatives to promote healthy lifestyle and prevent related diseases. They also proposed new dietary therapeutic approach to address some metabolic diseases from genera to particular.

My suggestions are:

·         The article is well composed and focused. However, its not too much progressive.

·         The novelty was not found in the MS neither the authors stated their hypothesis as they claimed in the abstract.

·         There should be a sunburst diagram, a correlation diagram as well to elucidate a unique “novel dietary therapeutic approach” or add prospective in the MS.

·         In my opinions, more of the data should be added and strong correlations amongst the metabolites, their sources, doses (daily intake), calories (approximate), mode of feed etc. should be added in this review for better presentation of the idea.

English is OK

Author Response

Dear reviewer, all comments were taken care of and are shown point by point.

Responses to the referees’ comments.

REVIEWER 2

The MS entitled “Bioactive chemicals in unprocessed foods and how they help prevent non-communicable diseases.” Was reviewed carefully. The MS describes the main health issues associated with poor diets and lacking bioactive compounds. Moreover, they also wrote about the nutritional alternatives to promote healthy lifestyle and prevent related diseases. They also proposed new dietary therapeutic approach to address some metabolic diseases from genera to particular.

My suggestions are:

  1. The article is well composed and focused. However, its not too much progressive. The title of the manuscript and the presentation of the content were adapted, thus strengthening the theories presented and the structure to maintain a clear and precise concordance.
  1. The novelty was not found in the MS neither the authors stated their hypothesis as they claimed in the abstract. The novelty focuses on gathering in a single document information on precise epidemiological data on the main Noncommunicable Diseases (NCDs) suffered in the world, followed by the close relationship of these conditions with the inflammatory process at the cellular and molecular level. In a third point, the nutraceutical compounds contained in foods of horticultural origin, grains and plants given their biological and beneficial properties in health are addressed; This being the main reason for proposing nutraceutical compounds in the diet as a preventive alternative and as an adjuvant in the treatment of NCDs.

Finally, the content of the abstract section was duly addressed and in accordance with the content of the manuscript here presented.

  1. There should be a sunburst diagram, a correlation diagram as well to elucidate a unique “novel dietary therapeutic approach” or add prospective in the MS. The NCDs information was rearranged in the corresponding sections, the source of origin of the bioactive compounds, such as fruits, vegetables, grains, was completed. In the conclusions section, proposals are made on the perspectives in this area.
  2. In my opinions, more of the data should be added and strong correlations amongst the metabolites, their sources, doses (daily intake), calories (approximate), mode of feed etc. should be added in this review for better presentation of the idea. The NCDs information was rearranged in the corresponding sections, the source of origin of the bioactive compounds, such as fruits, vegetables, grains, was completed. In the conclusions section, proposals are made on the perspectives in this area.

Comments on the Quality of English Language: English is OK

Reviewer 3 Report

This review work collects the literature regarding the link between bioactive compounds and health effects and in particular in relation to non-communicable diseases. The article contains a lot of information and an updated bibliography. some details should be improved with small revisions:

in particular, being a revision work on highly studied subjects and compounds, it is necessary to add the bibliographic references in the following points: in line 121, 312, 328.

-In the introduction, the last part (lines 53-61) can be moved and inserted the Methods paragraph.

Author Response

Dear reviewer, all comments were taken care of and are shown point by point.

Responses to the referees’ comments.

REVIEWER 3:

  1. This review work collects the literature regarding the link between bioactive compounds and health effects and in particular in relation to non-communicable diseases. The article contains a lot of information and an updated bibliography. some details should be improved with small revisions: This observation was addressed by updating information.
  2. in particular, being a revision work on highly studied subjects and compounds, it is necessary to add the bibliographic references in the following points: in line 121, 312, 328. This revision was heeded
  3. -In the introduction, the last part (lines 53-61) can be moved and inserted the Methods paragraph. The introduction was re-edited and the recommendation was heeded.

Round 2

Reviewer 1 Report

Comments resolved

Reviewer 2 Report

Dear Authors

Some of my suggestions have been addressed and it is Ok from my side.